# Assessing the Effect of Silicon Supply on Root Sulfur Uptake in S-Fed and S-Deprived *Brassica napus* L.

**DOI:** 10.3390/plants11121606

**Published:** 2022-06-18

**Authors:** Philippe Laîné, Raphaël Coquerel, Mustapha Arkoun, Jacques Trouverie, Philippe Etienne

**Affiliations:** 1Unicaen, INRAE, UMR 950 EVA, SF Normandie Végétal (FED4277), Normandie Université, 14000 Caen, France; philippe.laine@unicaen.fr (P.L.); raphael.coquerel@unicaen.fr (R.C.); jacques.trouverie@unicaen.fr (J.T.); 2Laboratoire de Nutrition Végétale, Agro Innovation International-TIMAC AGRO, 35400 Saint-Malo, France; mustapha.arkoun@roullier.com

**Keywords:** root sulfate transporters, S deficiency, silicon, sulfate influx, ^34^SO_4_^2−^, transcriptional regulation

## Abstract

Silicon (Si) is known to alleviate many nutritional stresses. However, in *Brassica napus*, which is a highly S-demanding species, the Si effect on S deficiency remains undocumented. The aim of this study was to assess whether Si alleviates the negative effects of S deficiency on *Brassica napus* and modulates root sulfate uptake capacity and S accumulation. For this, *Brassica napus* plants were cultivated with or without S and supplied or not supplied with Si. The effects of Si on S content, growth, expression of sulfate transporter genes (*BnaSultr1.1*; *BnaSultr1.2*) and sulfate transporters activity in roots were monitored. Si supply did not mitigate growth or S status alterations due to S deprivation but moderated the expression of *BnaSultr1.1* in S-deprived plants without affecting the activity of root sulfate transporters. The effects of Si on the amount of S taken-up and on S transporter gene expression were also evaluated after 72 h of S resupply. In S-deprived plants, S re-feeding led to a strong decrease in the expression of both S transporter genes as expected, except in Si-treated plants where *BnaSultr1.1* expression was maintained over time. This result is discussed in relation to the similar amount of S accumulated regardless of the Si treatment.

## 1. Introduction

Since the 1980s, the occurrence of sulfur deficiencies has been increasingly observed in field crops [1,2]. This low availability of S in soils is due to the combination of several factors including a significant decrease in S atmospheric emissions from anthropic activities, a decrease in the use of organic fertilizers and use of sulfur-free fertilizers associated with intensive management practices [3,4]. In this context, plants are increasingly exposed to S deficiencies that are particularly detrimental for plants with high sulfur requirements for growth and seed production. This is notable in *Brassica napus* L. whose seed yield and seed quality are negatively affected by S deficiency [5,6]. Phenotypically, S deficiency usually leads to a decrease in growth, a yellowing of young leaves, and production of anthocyanin in mature leaves [7]. Moreover, some studies have reported that genes encoding SO_4_^2−^ transporters, especially those encoding *BnaSultr1.1* and *BnaSultr1.2* (both of which are high-affinity transporters mainly involved in root sulfate uptake) were strongly up-regulated in roots of oilseed rape subjected to S-deficiency [8,9]. Studies performed in numerous plant species have reported that because S is a constituent of cysteine and methionine, S deficiency leads to a disruption of many aspects of plant metabolism including S and N metabolism [8,10,11]. In the same way, because S is also required for the synthesis of phytohormones [12] and non-enzymatic antioxidant compounds such glutathione and S-adenosine methionine [13,14,15], S deficiency is usually associated with an alteration of growth and defense mechanisms in plants facing biotic or biotic stresses [16].

Interestingly, the addition of Si, especially to crops under biotic and abiotic stress conditions, improves their resistance and the quality and yield of the crops harvested [17,18,19]. Although most work has been conducted on plants with high Si accumulation such as rice [20], several studies have also shown that Si supply can be beneficial in plants with low Si accumulation, such as *Brassica napus*. In general, one of the common defense mechanisms triggered by Si to mitigate biotic and abiotic stresses is the detoxification of reactive oxygen species (ROS) [21]. For example, in cucumber roots infected with *Pythium* spp., Cherif et al. [22] reported that Si supply induced peroxidase and polyphenoloxidase activities. In particular, in *Brassica napus* and *Brassica juncea* exposed to salinity or drought, Si was shown to upregulate a wide range of non-enzymatic (ascorbate and glutathione pools) and enzymatic antioxidants activities [23,24,25,26]. In addition to alleviating drought and salinity stresses, many studies report that Si supply is also beneficial in helping plants overcome main macronutrient (N, P, K) deficiencies. Two main mechanisms for Si-mediated deficiency mitigation have been highlighted: increased nutrient uptake in roots, and improved nutrient utilization in plant tissues (for review: [27]). For example, in rice cultivated under N starvation, Si has been reported to enhance the rate of N assimilation [28] and to improve the performance of ammonium transporters (OsAMT), but it does not induce *OsAMT* gene expression [29]. In N-deprived *Brassica napus* plants, Si supply induced the expression of genes encoding root nitrate transporters (*BnaNTR1.1* and *BnaNTR2.1*) and led to an improved nitrogen uptake and growth recovery when the plants were resupplied with N for 12 days [30]. Unlike other macronutrients, only a few studies have dealt with Si and S interactions [27,31] and the early results seem to depend on the plant species. Indeed, Buchelt, et al. [32] reported no effect of Si on S uptake accumulation in forage crops, while other authors have indicated that Si supply results in a decreased expression of genes encoding sulfur transporters (*OsSultr1.1* and *OsSultr2.1*) [33] and a consistently decreased S accumulation in shoots [33,34] in rice or barley subjected to S deficiency. For *Brassica napus*, which unlike the above-mentioned species does not accumulate Si and has a high demand for sulfur, there are no studies to date on S-Si interactions [31].

In this context, the aims of this study were to assess whether a Si supply (i) alleviated stress associated with S deficiency in *Brassica napus* and/or (ii) modulated the root sulfate transporters activity and S accumulation in plants. For this, the effect of Si supply on growth, expression of root sulfate transporter genes (*Bnasultr1.1* and *Bnasultr1.2*) were monitored. To evaluate the activity of sulfate transporters in plants fed with (+S) or without (–S) sulfate during 21 days (D21), sulfate influx was measured at D21 using short-term labelling with ^34^SO_4_^2−^. Finally, the effect of Si supply on modulating the expression of S transporters observed in this study and its consequences on S uptake during a 72 h S re-feeding are discussed.

## 2. Results

### 2.1. Effect of S Deficiency Associated or Not Associated with Si Supply on Brassica napus Growth

For each treatment, a representative image of the plants cultivated under hydroponic conditions is shown in Figure 1A. Visually, regardless of the Si treatment (+Si or –Si), it can be observed that plants deprived of S (–S) for 21 days were smaller than plants well supplied with S (+S). In agreement with the images in Figure 1A, the total dry weight of -S plants was significantly reduced compared to +S plants. Indeed, between D0 and D21, the total dry weight of plants evolved from 0.96 ± 0.01 g.plant^−1^ to 11.51 ± 1.22, 12.46 ± 1.18, 8.19 ± 0.71 and 8.79 ± 1.17 g.plant^−1^ for +S–Si, +S+Si, –S–Si and –S+Si, respectively (Figure 1B). This difference was due to a tendency for the shoot biomass to decrease and especially to a significant decrease in the dry weight of the roots of -S plants compared to the +S plants. Irrespective of the treatment, the shoot/root ratio at D21 did not change significantly (4.11 ± 0.49, 4.71 ± 0.46, 4.80 ± 0.33 and 4.83 ± 0.57 for +S–Si, +S+Si, –S–Si and –S+Si, respectively). In addition, Figure 1B shows that whatever the S supply (+S or –S), Si treatment had no significant effect on shoot and root dry weight.

### 2.2. Leaf Chlorophyll and Anthocyanin Indices

In agreement with several previous studies [35,36], an optical sensor (Multiplex^®^) was used to evaluate the chlorophyll and anthocyanin indices of *Brassica napus* leaves subjected to the different treatments (+S–Si; +S+Si, –S–Si, and –S+Si) for 21 days (Figure 2A,B). Regardless of the treatment, the chlorophyll (SFR_R and SFR_G) index remained constant (around 5) (Figure 2A). Figure 2B shows that when plants were fed with S (+S), silicon supply (+Si) there was no significant effect on the anthocyanin index of aerial parts. In contrast, in S-deprived plants (–S), silicon supply (+Si) significantly increased the anthocyanin index of aerial parts (from 0.005 ± 0.004 to 0.021 ± 0.003 in –S–Si and –S+Si plants, respectively). Moreover, our results indicated that a typical increase in the anthocyanin index in response to S deprivation was significant only when plants were fed with Si (+Si). This result correlates with the purplish coloration of leaves that can be observed in the –S+Si plants in Figure 1A.

### 2.3. Contents of Sulfur and Silicon in Plants

As expected, whatever the Si treatment (+Si or –Si), the S content in S-deprived whole plants at D21 was significantly lower (around 10-fold) than in S-fed plants (+S) and comparable to that in plants at D0 (Figure 3A). However, the distribution of the S between roots (25%) and shoots (75%) did not change, irrespective of the treatment. As is usually reported for Si content, despite the use of nutrient solution without Si, -Si plants showed an increase in the content of Si between D0 and D21 (Figure 3B). Nevertheless, addition of sodium metasilicate (Na_2_SiO_3_) in the nutrient solution for 21 days led to a significant increase in the Si content in whole plants compared to plants cultivated without Si (–Si), regardless of the S supply (+S or –S). Furthermore, among the +Si plants, the content of Si in the S-fed plants (+S) was significantly higher than in S-deprived plants (–S). Interestingly, this increase in the content of Si was mainly due to Si accumulation in roots, for example, from 2.5 ± 0.2 to 11.6 ± 1.5 mg plant^−1^ between +S–Si and +S+Si plants (around 4.5-fold), whereas the Si content was less increased (around 1.5-fold) in leaves of the same plants (Figure 3B).

### 2.4. Relative Gene Expression of BnaSultr1.1 and BnaSutr1.2 in Roots

In S-fed plants (+S), the relative expression of genes encoding the two sulfate transporters, *BnaSutr1.1* and *BnaSultr1.2*, was not modified in response to Si supply. Compared to +S plants, S deprivation led to a strong induction of the genes encoding both transporters (Figure 4A,B). For example, the expression of *BnaSultr1.2* in S-deprived plants was increased 30-fold on average, regardless of the Si supply (–Si or +Si) (Figure 4B). Interestingly, even though the expression of *BnaSultr1.1* was also very strongly induced in response to S deprivation, it is notable that this induction was around two fold lower (75 vs. 160) in –S+Si plants than in –S–Si plants.

### 2.5. Root Sulfate Transporters Activity

The activity of sulfate transporters in plants fed with (+S) or without (–S) sulfate during 21 days (D21) was determined by measuring sulfate influx at D21 using short-term labelling with ^34^SO_4_^2−^ (Figure 5A). In S-fed plants (+S), the root sulfate transporter activity reached around 6.5 µmol SO_4_^2−^ g^−1^ of root DW h^−1^, irrespective of the Si treatment (–Si or +Si). In S-deprived plants, the activity of root sulfate transporters is stronger than in S-fed plants and achieves 15 µmol SO_4_^2−^ g^−1^ of root DW in both –S–Si and –S+Si plants (Figure 5A). Interestingly, despite a negative effect of Si supply on expression of genes encoding root transporters (especially *BnaSultr1.1*) in S-deprived plants, the root sulfate transporters activity of the –S–Si and –S+Si plants were similar. To explore this further, the total S taken up by plants was determined after 3 days of S resupply (+S_72h_). The results indicated that the S amount taken up during the 72 h of S resupply by previously S-fed plants (+S) was similar, regardless of the Si treatment (around 5 mg S plant^−1^ in both (+S–Si) + S_72h_ and (+S+Si) + S_72h_ plants) (Figure 5B). In plants previously subjected to S deprivation (–S), the S amount taken up was increased much more strongly than in +S plants, and reached around 25 mg S plant^−1^ in both +Si ((-S+Si) + S_72h_) and –Si ((–S–Si) + S_72h_) plants. Consistent with the results discussed above, compared to 21 day, this strong sulfate uptake led to an increase in S contents in both plant compartments in (–S–Si) + S_72h_ and (–S+Si) + S_72h_ plants (2- and 3-fold in roots and shoots, respectively), while the S contents in roots and shoots of +S plants remained similar (Figure 5C).

### 2.6. Relative Expression of Genes Encoding Root Sulfate Transporters in Plants Resupplied with S

Compared to D21, the relative expression of the genes encoding the BnaSultr1.1 and BnaSultr1.2 sulfate transporters in +S plants remained similar at D21 + 72 h (+S–Si) + S_72h_ and (+S+Si) + S_72h_ (Figure 6). Compared to D21, in previously S-deprived plants without Si ((+S–Si) + S_72h_), the S resupply for three days (D21 + 72 h) resulted in a significant down-regulation of expression of both genes (around 3-fold) (Figure 6). In previously S-deprived plants fed with Si ((+S+Si) + S_72h_), the S resupply also led to a down regulation of *BnaSultr1.2* (Figure 6B) while expression of *BnaSutr1.1* remained high and at the same level as at D21 (Figure 6A).

## 3. Discussion

The aim of this study was to evaluate the effect of Si supply on growth and sulfate root transporters activity in *Brassica napus* fed with or without sulfate. After 21 days, plant biomasses were significantly lower in the S-deprived plants than in the S-fed plants (Figure 1), as reported in previous studies performed on *Brassica napus* grown under similar hydroponic conditions and the same duration of S deficiency [37]. In the current study, despite the addition of Si being accompanied by an increase in the content of Si in the plant (Figure 3B), there was no impact on plant growth (Figure 1) regardless of the sulfur treatment (–S or +S). These results are in agreement with previous studies that generally reported minor or no beneficial effects of Si on biomass production in unstressed plants [18,38], nor on plants subjected to severe deficiencies in essential macronutrients such as N, P and K [39,40]. A previous study showed that one of the beneficial effects of Si supply was to delay chlorophyll degradation in *Brassica napus* under N deficiency [30]. Therefore, chlorophyll content was monitored to assess whether Si supply had a similar effect on plants grown in S deficiency. As shown in Figure 2A, such an effect was not observed on the chlorophyll indices in either +S plants, which was expected, or in –S plants. The lack of effect of Si on S-deprived plants can also be explained because a delay in chlorophyll degradation by Si is usually reported when senescence is induced by nutritional stress [30,41] and because S deprivation is instead known to not induce or even sometimes to delay senescence [42]. In contrast to chlorophyll, the leaf anthocyanin content was strongly increased when S-deprived plants were fed with Si compared to –S–Si plants (Figure 1A and Figure 2B). This result agrees with earlier studies reporting that Si treatment leads to an increase in anthocyanin content in several plant species [43,44,45]. This accumulation of anthocyanins under Si treatment could be an advantage in coping with biotic and abiotic stresses due to the anthocyanins’ non-enzymatic antioxidant properties, which is particularly important under –S conditions. Indeed, whatever the Si supply, the low content of S in -S plants (Figure 3A) suggested, as classically reported, that S compounds and specifically those with antioxidant properties (such as glutathione) might be lacking. This is especially true when plants are supplied with Si because it was demonstrated that Si further lowered the low glutathione content in S-deprived plants [34].

As already reported in many studies [8,9,11], sulfur deficiency led to a strong induction of the expression of genes encoding two transporters (*BnaSultr1.1* and *BnaSutr1.2*; Figure 4A,B) that are mainly involved in sulfate uptake by roots [46]. Nevertheless, it can be highlighted that in –S plants, supplying Si was associated with a lower induction of *BnaSutr1.1* gene expression than in –S–Si plants (Figure 4A). This result is consistent with other work performed in rice where expression of the *Sultr1.1* gene was lower when S-deprived plants were cultivated in the presence of Si compared to S-deprived plants grown without Si supply [33]. However, this result contrasts with what is observed in plants deprived of N or P, where silicon supply induces the root expression of transporters involved in the uptake of these nutrients [30,47]. Because sulfur is involved in the maintenance of redox status, it is possible that the lower induction of *BnaSultr1.1* in the presence of Si could be due to a lower need for sulfur as Si is known to have a beneficial effect on the redox status of plants [30,48]. This beneficial effect could be a consequence of an increase in sulfur-free antioxidant metabolites such as the anthocyanin accumulation described previously in –S+Si plants (Figure 2B) but also of the induction of many ROS detoxifying enzymes by Si, as reported in many studies (for review: [49]). This assumption is reinforced by the lower hydrogen peroxide (H_2_O_2_) and malondialdehyde (MDA) contents in shoots of –S+Si plants than in –S–Si plants observed in our study (Appendix A). Due to the S-transporter gene expression induction being observed in the complete absence of S supply, the contents of sulfur observed at D21 were not significantly different from D0 (from 7 to 10 mg plant^−1^; Figure 3A) and do not allow to evaluate the effect of Si on the actual sulfur uptake during this period. To overcome this, the instantaneous root sulfate transporters activity of plants was measured at D21 using ^34^S-labelled sulfate (Figure 5A). Despite a strong induction in S-deprived plants and contrasting levels of induction according to the Si level (160- and 75-fold for *BnaSultr1.1* and 35- and 26-fold for *BnaSultr1.2* in –S–Si and –S+Si plants, respectively), the variable expression of the genes encoding sulfate transporters results in a sulfate transport activity that was only double the levels in the +S plants. This apparent mismatch between the high level of expression of genes encoding S transporters and the capacity to take up S suggests a limitation in the ability of S-deprived plants to synthesize (post-transcriptional regulation) or operate sulfate transporters (metabolic perturbation). For example, this limitation of transporter synthesis could be a consequence of the numerous metabolic disturbances and more particularly, the decrease in the biosynthesis of sulfur amino acids and proteins usually observed in S-deprived plants [11,50]. A previous study showed that S resupply was accompanied by a strong increase in de novo synthesis of amino acids and proteins [9]. If transporter synthesis is indeed prevented by S availability, then it cannot be excluded that the higher expression of *BnaSultr1.1* in -S-Si plants (compared to –S+Si plants) under S resupply may result in higher transporter synthesis that might be associated with higher sulfate uptake (and *vice versa*) upon S resupply. To check this hypothesis, the amount of sulfur taken up as well as the S contents were evaluated in plants previously deprived of sulfur and re-fed with sulfur for 72 h (Figure 5B,C). After 72 h of S re-supply, the increase in tissue contents of S-replenished plants indicated that in addition to using the absorbed S for growth (+3 g in 3 days, data not shown), such plants were again able to store S in their tissues and use it locally to sustain S metabolism. Moreover, even though the cumulative quantity of S absorbed in 72 h was significantly higher in –S plants than in +S plants, it is interesting to note that the amount of sulfur absorbed during this period was independent of the Si treatment. This result parallels work performed on forage crops, which demonstrated that Si supply did not affect uptake and accumulation of S [32]. Furthermore, after 72 h of S resupply in Si-treated plants we observed a repression of the gene expression of both of the sulfate transporters (from 160- to 46-fold for *Bnasultr1.1* and from 37- to 10-fold for *BnaSultr1.2* between D21 and D21 + 72 h; Figure 6) as typically observed in such situations [9,51]. More surprisingly, our study highlights no repression of the *BnaSultr1.1* gene after 72 h of S resupply in Si-treated plants (Figure 6A). Under S starvation, it has been shown that the genes encoding the sulfate transporters *Sultr1.1* and *Sultr1.2* are induced by a low sulfur assimilation capacity, which leads in particular to a non-use of OAS and to its accumulation [10,52]. In contrast, when sulfur is resupplied, the decrease in OAS content following the resumption of sulfur metabolism leads to a repression of these root transporters [16]. During the S resupply performed in our study, the absence of repression of *BnaSultr1.1* gene expression in S-deprived and Si-treated plants alone suggested that silicon might interact directly or not with the regulatory pathway described above. In any case, the different expression patterns of *BnaSultr1.1* that manifested in the presence or absence of Si supply (i.e., a strong expression at D21 followed by a strong repression at D21 + 72 h in –S–Si plants versus a more moderate expression (×75) at D21 but maintained over time (D21 + 72) in –S+Si plants) might provide an explanation for the similar amounts of sulfur taken-up during the 72 h S re-supply in plants previously deprived of S, irrespective of the Si treatment.

In conclusion, according to a recent review [31], the current study is only the third to investigate S-Si interactions and the first to explore the effect of Si supply on S uptake in *Brassica napus* L., a field crop of substantial agronomic interest with high sulfur requirements and low Si accumulation. This study reveals that although Si supply does not lead to changes in sulfur status, a significant effect on the expression of genes encoding sulfate transporters has been observed. This apparent discrepancy raises questions about the impact of Si on the transcriptional and post-transcriptional regulatory pathways of S transporters in *Brassica napus* cultivated under S deficiency. Future investigations combining comparative omics approaches should be undertaken to decipher the mechanisms involved in these regulatory pathways.

## 4. Materials and Methods

### 4.1. Plant Growth Conditions and Experimental Design

The experimental design is summarized in Figure 7. In a greenhouse, seeds of *Brassica napus* L. var. “Citizzen” were germinated on perlite over deionized water for four days in the dark. Then seedlings were transferred to natural light conditions and supplied with nutrient solution for two weeks containing: KNO₃ (1 mM), KH₂PO₄ (0.25 mM), KCl (1 mM), CaCl_2_ (3 mM), MgSO₄ (0.5 mM), EDTA-2NaFe (0.2 mM), H_3_BO_3_ (14 µM), MnSO_4_ (5 µM), ZnSO_4_ (3 µM), CuSO_4_ (0.7 µM), (NH_4_)_6_Mo_7_O_24_ (0.7 µM), CoCl_2_ (0.1 µM). This nutrient solution classically used for *Brassica napus* cultivated in hydropnic conditions is derived from Hoagland nutrient solution and results in mineral levels in plant tissues comparable to those observed in field-grown plants [37,53,54]. Just after first leaf emergence (i.e., after 2 weeks), seedlings were transplanted for one week into a plastic tank (20 L) containing the nutrient solution described above. Natural light was supplied by high pressure sodium lamps (Philips, MASTER GreenPower T400W) with photosynthetically active radiation of 450 μmol photons m^−2^ s^−1^ at canopy height. At the emergence of the fourth leaf (D0), plants were separated into four batches: the first two batches corresponded to plants maintained on S (+S; 500 µM) and supplied (+S+Si) or not (+S–Si) with 1.7 mM silicon for 21 days (D21), the maximal concentration for which Si remained soluble [55]. The nutrient solution described above with the addition of sodium metasilicate (Na_2_SiO_3_) to reach a final concentration of 1.7 mM Si was provided to +S+Si plants. The other two batches corresponded to plants deficient in S (–S; 0 µM) and supplied (+Si) or not (–Si) with silicon (1.7 mM). For these plants, nutrient solution was modified as follows to suppress S: KNO₃ (1 mM), KH₂PO₄ (0.25 mM), KCl (1 mM), CaCl_2_ (3 mM), MgCl_2_ (0.5 mM), EDTA-2NaFe (0.2 mM), H_3_BO_3_ (14 µM), MnCl_2_ (5 µM), ZnCl_2_ (3 µM), CuCl_2_ (0.7 µM), (NH_4_)_6_Mo_7_O_24_ (0.7 µM), CoCl_2_ (0.1 µM). Whatever the S treatment, Si-deprived plants were supplied with NaCl (final concentration: 3.4 mM) to compensate the sodium supplied by Si treatment. From D21, irrespective of the treatment, the plants were grown for 72 h (D21 + 72 h) with S (+S_72h_; 500 µM) using the initial nutrient solution. During plant cultivation, nutrient solutions were renewed every three days and their pH was adjusted to 5.8 ± 0.2. As described in several studies [35,36,56], the chlorophyll and anthocyanin contents were assessed in aerial parts at day 21 (D21) with an optical sensor system (Multiplex^®^ fluorometer, Force A, Orsay, France). Plants were harvested at D0, D21 and D21 + 72 h (Figure 1). At each harvest, shoots and roots were separated and then frozen in liquid N and stored at −80 °C for further analysis. An aliquot of each tissue was dried in an oven (60 °C) for dry weight (DW) determination and elemental analysis.

### 4.2. Measurement of Sulfate Influx

The root sulfate influx was performed at D21 by using a short-term sulfate uptake labeling experiment (^34^SO_4_^2−^). Briefly, roots were washed twice for 1 min in a solution of Ca(NO_3_)_2_ (1 mM) before immersion for 5 min in a solution of 200 μM Na_2_^34^SO_4_^2−^ (90% of atom excess) to study root sulfate influx in plants grown with or without S and Si. Roots were then rinsed twice in a solution of Ca(NO_3_)_2_ (1 mM) at 4 °C for 1 min to stop the SO_4_^2^ uptake. Roots and shoots were separated and weighed before drying in an oven (60 °C) for dry weight (DW) determination. Total S and ^34^S contents in roots and shoots were determined with 6 mg of fine powder placed in tin capsules before analysis in an isotope-ratio mass spectrometer (Delta V Advantage, ThermoFisher, Bremen, Germany) linked to a C/N/S analyzer (EA3000, Euro Vector, Milan, Italy). The value of the isotope abundance (A%) given by the IRMS in sample or natural standard is calculated as:A%=100×S34/S34+S32

Then, the content of ^34^S (Q^34^S) in each sample was calculated as following:Q^34^S = (A%_sample_ − A%_natural standard_) × QS/100
where QS represents the total sulfur contents and A%_natural standard_ represents the natural abundance of ^34^S, i.e., 4.2549%.

Finally, sulfate influx was expressed as the total ^34^S content in whole plant (Q^34^S) per gram of dry roots and per hour.

### 4.3. Determination of Total S and Si Contents

In roots and shoots from unlabeled plants, Si and S contents were quantified with approximately 1 g of dry weight powder analyzed with an X-ray-fluorescence spectrometer (XEPOS, Ametek, Berwyn, PA, United States) using calibration curves obtained from international standards with known contents.

### 4.4. Extraction and Quantification of RNAs, Reverse Transcription (RT) and Q-PCR Analysis

At D21 and D21 + 72 h, total RNAs were extracted from frozen root tissue previously ground in a mortar containing liquid nitrogen. 200 mg of powder was suspended in 750 µL extraction buffer (100 mM LiCl, 100 mM TRIS, 10 mM EDTA, 1% SDS (*w*/*v*), pH 8) and 750 µL of hot phenol (80 °C, pH 4). After vortexing for 40 s, 750 µL of chloroform:isoamylalcohol (24/1, *v*/*v*) was added. After centrifugation of the mixture (15,000 g, 5 min, 4 °C) the supernatant was collected and 750 µL of 4 M LiCl solution (*w*/*v*) was added before overnight incubation at 4 °C. After centrifugation at 15,000 g for 20 min at 4 °C, the pellet containing total RNAs was resuspended with 100 µL of sterile water before purification with an RNeasy Mini Kit according to the manufacturer’s protocol (Qiagen, Courtaboeuf, France). Total RNA was quantified by spectrophotometry at 260 nm (BioPhotometer, Eppendorf, Le Pecq, France) before reverse transcription (RT) step.

For RT, 1 µg of total RNAs was converted to cDNA with an iScript cDNA synthesis kit according to the manufacturer’s protocol (Bio-Rad, Marne-la-Coquette, France). Relative expression of genes encoding sulfate transporters (*BnaSultr1.1*; *accession number*: AJ416460 and *BnaSultr1.2*; *accession number*: AJ311388) was monitored in roots using quantitative polymerase chain reaction (q-PCR). Q-PCR amplifications were performed using specific primers for each housekeeping gene (*EF1-α* (*accession number:* DQ312264); forward: 5′-gcctggtatggttgtgacct-3′, reverse: 5′-gaagttagcagcacccttgg-3′ and *18S rRNA (accession number:* X16077); forward: 5′-cggataaccgtagtaattctag-3′, 5′-reverse: gtactcattccaattaccagac-3′) and target genes (*BnaSultr1.1: forward:* 5′-agatattgcgatcggaccag-3′, reverse: 5′-gaaaacgccagcaaagaaag-3′ and *BnaSultr1.2:* 5′-ggtgtagtcgctggaatggt-3′, 5′-aacggagtgaggaagagcaa-3′).

q-PCRs were performed with 4 µL of 100x diluted cDNA, 500 nM of primers, and 1x SYBR Green PCR Master Mix (BioRad, Marne–la–Coquette, France) in a thermocycler (CFX96 Real Time System, Bio–Rad, Marne–la–Coquette, France). A program composed of an activation step at 95 °C for 3 min and 40 cycles of a denaturing step at 95 °C for 15 s followed by an annealing and extending step at 60 °C for 40 s, was used. For each pair of primers, PCR efficiency was around 100% and the specificity of PCR amplification was monitored though the presence of a single peak in the melting curves after the q-PCR. In addition, the q-PCR product was sequenced to confirm that the correct amplicon was produced from each pair of primers (Eurofins, Ebersberg, Germany). For each sample, the subsequent q-PCRs were performed in four biological replicates (n = 4). At each harvest time (D21 and D21 + 72 h), the relative expression of the genes in each sample was compared with the referent sample (corresponding to the +S–Si treatment) using the delta Ct (ΔΔCt) method and the following equation:

Relative expression = 2^−ΔΔCt^, with ΔΔCt = ΔCt_sample_ − ΔCt_referent_ and with ΔCt = Ct_target gene_ − Ct_housekeeping gene_, where Ct refers to the threshold cycle determined for each gene (housekeeping or target) in the exponential phase of PCR amplification. Using this analysis method, the relative expression of the target gene in the referent sample (+S–Si) was equal to 1 [57].

### 4.5. Statistical Analysis

The experiment was performed with four independent replicates. Thus, all data are indicated as the mean ± S.E (*n* = 4). Statistical analyses were performed using R software (version 4.2.0: R Core Team, 2022) and Rcmdr (version 2.7-2). After normality analysis (Shapiro-Wilk test), data were analyzed using analysis of variance (ANOVA) and mean values were compared using Tukey’s HSD *post-hoc* test. For gene expression and S contents (Figure 6 and Figure 7), significant differences between data at D21 and D21 + 72 h were determined using Student’s *t*-test (*p* < 0.05).

## Figures and Tables

**Figure 1 plants-11-01606-f001:**
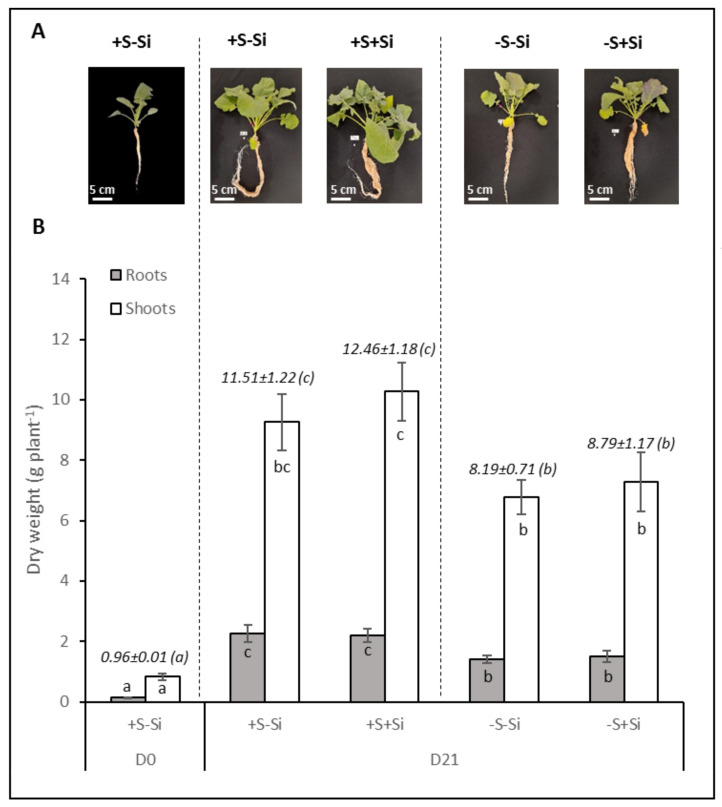
Representative pictures (**A**) and dry weights (**B**) of roots and shoots of *Brassica napus* L. plants cultivated in hydroponic conditions for 21 days (from D0 to D21) with (+S; 500 µM) or without sulfur (–S; 0 µM) and with (+Si; 1.7 mM) or without silicon (-Si) supply. Italic values indicate the whole plant biomass. Data are means ± SE (n = 4). Different Latin letters indicate that the mean dry weight of whole plant or of each plant compartment (roots or shoots) are significantly different (*p* < 0.05).

**Figure 2 plants-11-01606-f002:**
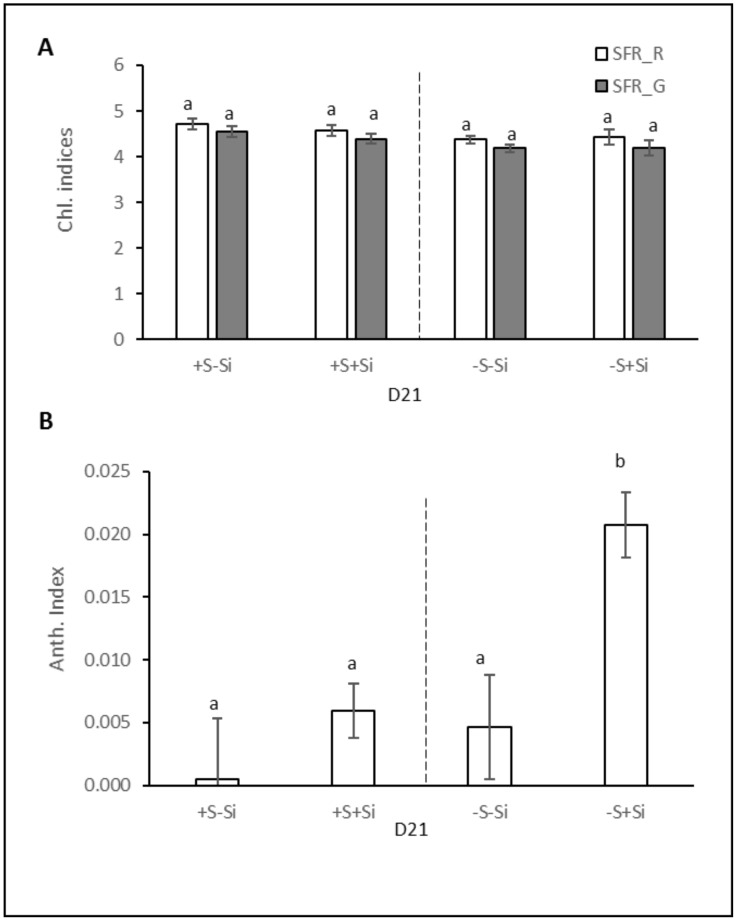
Chlorophyll (Chl., **A**) and anthocyanin (Anth., **B**) indices measured with an optical sensor (Multiplex) in shoots of *Brassica napus* L. cultivated in hydroponic conditions for 21 days (D21) with (+S; 500 µM) or without sulfur (–S; 0 µM) concentrations and with (+Si; 1.7 mM) or without silicon (-Si) supply. Data are means ± SE (n = 4). Different lowercase letters indicate significant difference in the indices (*p* < 0.05).

**Figure 3 plants-11-01606-f003:**
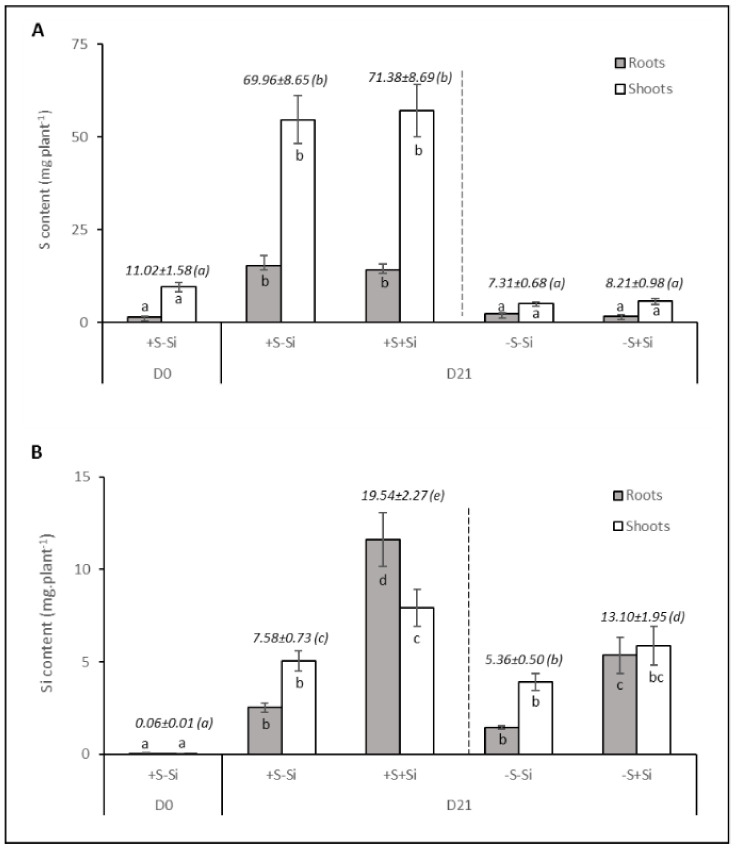
Roots, shoots and total contents of S (**A**) and Si (**B**) in *Brassica napus* L. cultivated in hydroponic conditions for 21 days (21D) with (+S; 500 µM) or without sulfur (–S; 0 µM) concentrations and with (+Si; 1.7 mM) or without silicon (–Si) supply. Italic values indicate the content S or Si in whole plant (mg plant^−1^). Data are means ± SE (n = 4). Different Latin letters indicate that the S or Si content in whole plant or in each plant compartment (roots or shoots) are significantly different (*p* < 0.05).

**Figure 4 plants-11-01606-f004:**
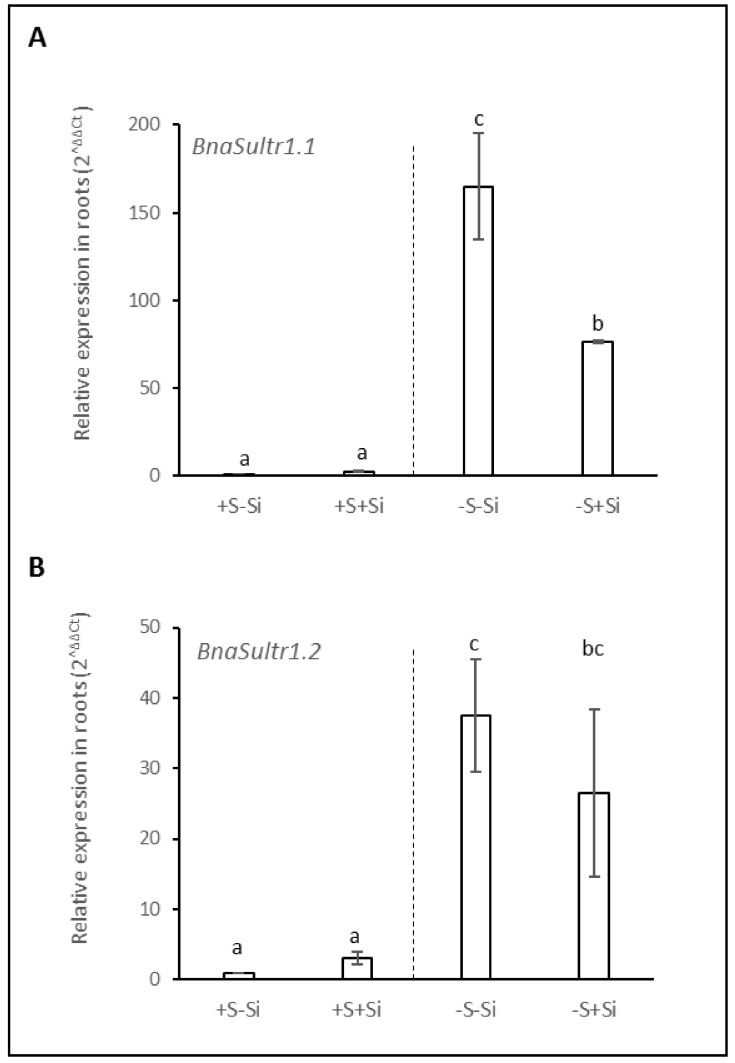
Relative expression of genes encoding the sulfate transporters *BnaSultr1.1* (**A**) and *BnaSultr1.2* (**B**) in roots of *Brassica napus* L. cultivated in hydroponic conditions for 21 days with (+S; 500 µM) or without sulfur (–S; 0 µM) concentrations and with (+Si; 1.7 mM) or without silicon (–Si) supply. Data are means ± SE (n = 4). Different lowercase letters indicate that means are significantly different (*p* < 0.05).

**Figure 5 plants-11-01606-f005:**
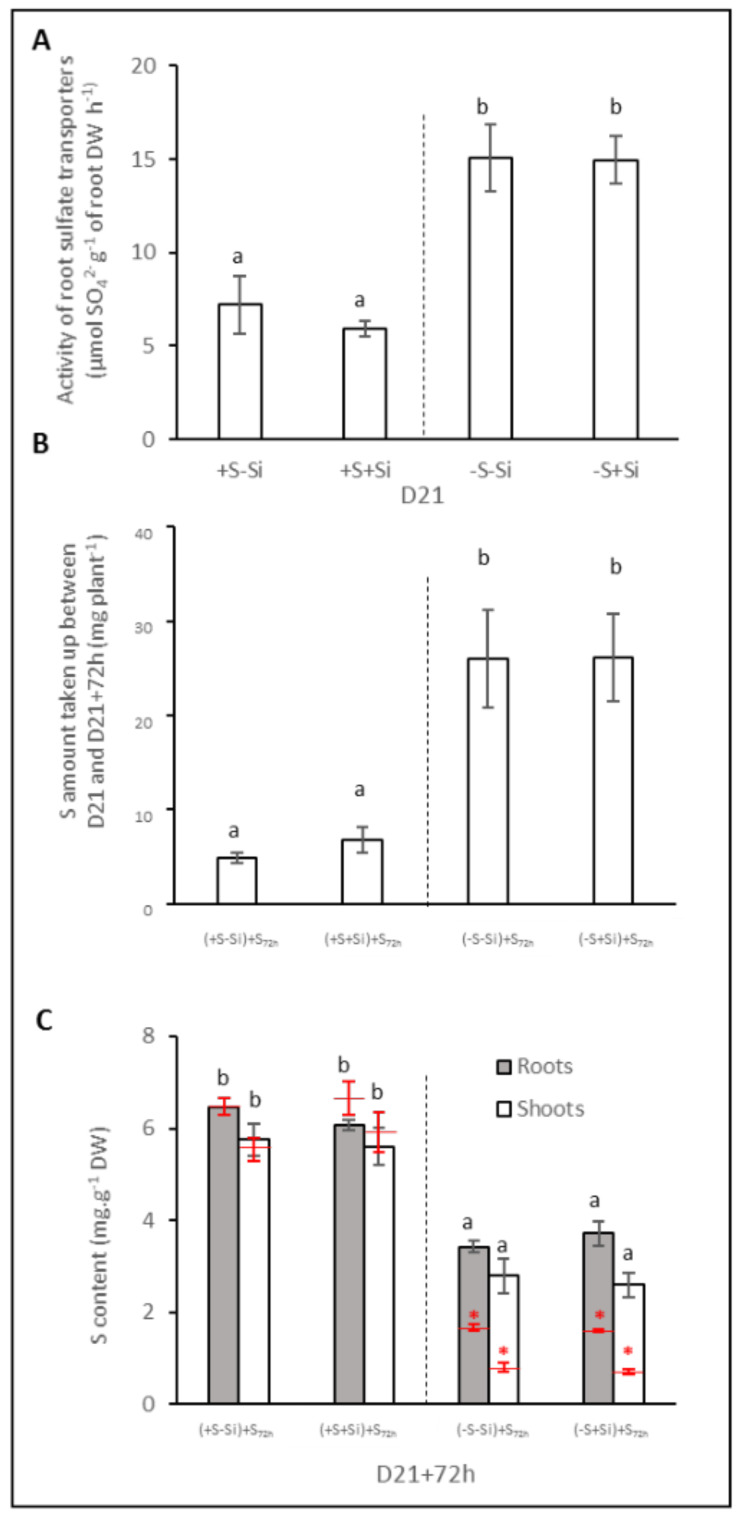
Activity of root sulfate transporters of *Brassica napus* L. at D21 (**A**), the S amount taken up between D21 and D21 + 72 h (**B**), and the S content in the root and shoot compartments of plants at D21 + 72 h (**C**). Plants were previously cultivated in hydroponic conditions for 21 days with (+S; 500 µM) or without sulfur (–S; 0 µM) concentrations and with (+Si; 1.7 mM) or without Si (–Si) supply and then were resupplied with S for 72 h (+S_72h_; 500 µM). Data are means ± SE (n = 4). Different lowercase letters indicate that the means are significantly different (*p* < 0.05). (**C**) for each treatment, red marks indicate the mean (±SE n = 4) of S content in the plant compartments at 21 days. For each treatment, the red asterisk indicates significant differences (*p* < 0.05) between the S amounts at D21 and at D21 + 72 h.

**Figure 6 plants-11-01606-f006:**
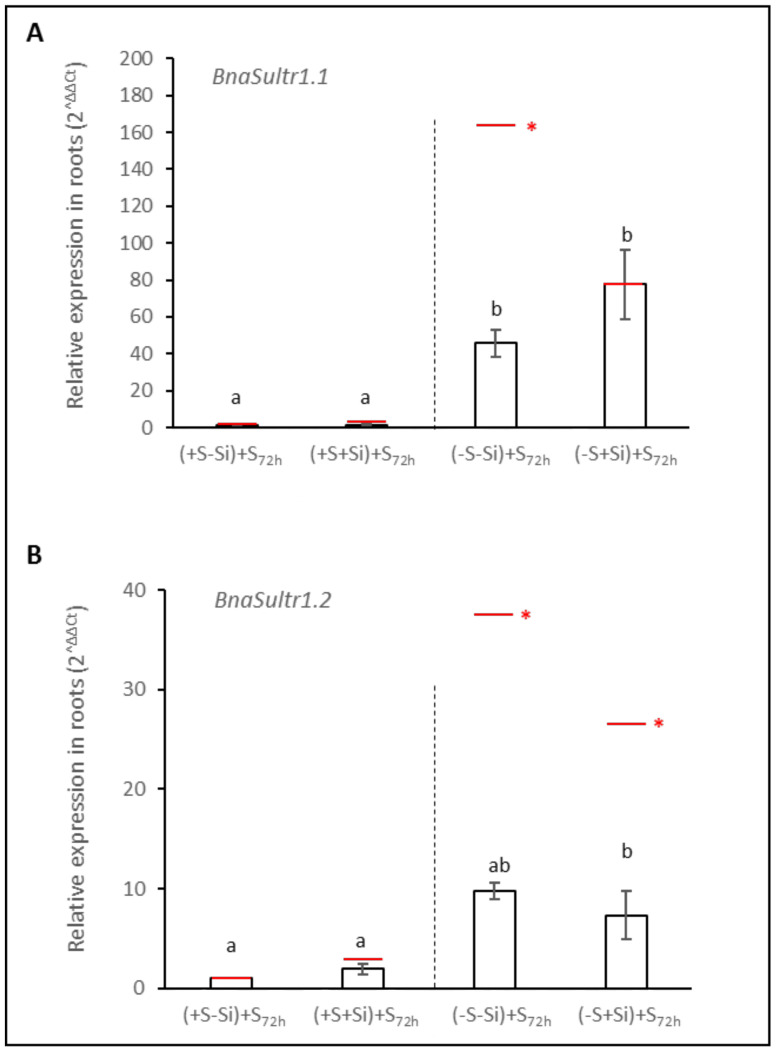
Relative expression of genes encoding the sulfate transporters *BnaSultr1.1* (**A**) and *BnaSultr1.2* (**B**) in roots of *Brassica napus* L. at D21 + 72 h: plants were previously cultivated in hydroponic conditions for 21 days with (+S; 500 µM) or without sulfur (–S; 0 µM) concentrations and with (+Si; 1.7 mM) or without Si supply (–Si) and then were resupplied with S for 72 h (+S_72h_; 500 µM). Data are means ± SE (n = 4). Different lowercase letters indicate that means are significantly different (*p* < 0.05). For each treatment, red marks represent the relative expression of each gene observed in roots at 21 days (Figure 5). For each treatment and each gene, asterisks indicate significant differences (*p* < 0.05) between the relative expression at D21 and at D21 + 72 h.

**Figure 7 plants-11-01606-f007:**
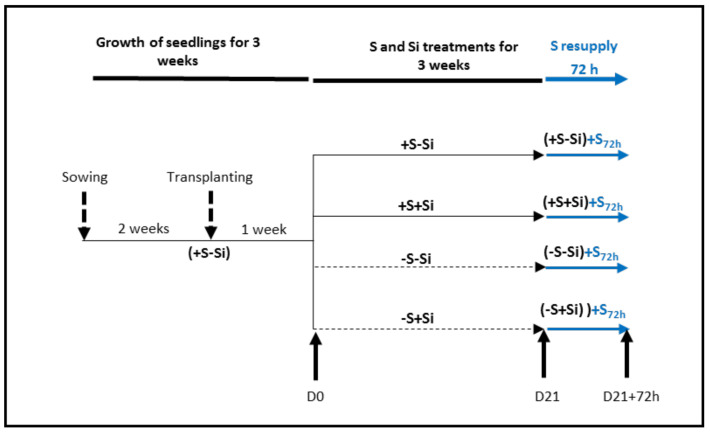
Experimental design used to study the effect of silicon supply on *Brassica napus* L. grown with or without sulfur for 21 days followed by S supplementation for 72 h. After three weeks of cultivation with sulfur (+S; 500 µM) (D0), plants were separated into four batches: the first two batches (filled black arrow) corresponded to plants that continued to be fed with S (+S; 500 µM) and supplied (+Si; 1.7 mM) or not (–Si) with silicon for 21 days (D21). The other two batches (dotted black arrow) corresponded to plants deficient in S (–S) and supplied (+Si; 1.7 mM) or not (–Si) with silicon. From D21, irrespective of the treatment, all plants were resupplied for 72 h (D21 + 72 h) with S (+S_72h_; 500 µM) (filled blue arrow).

## Data Availability

Not applicable.

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
