# Peer review of "Assessing the Effect of Silicon Supply on Root Sulfur Uptake in S-Fed and S-Deprived Brassica napus L."

_plants, 2022, doi:10.3390/plants11121606_

Round 1

Reviewer 1 Report

The articleAssessing the effect of silicon supply on root sulfur uptake in 2 S-fed or S-deprived Brassica napus L.” described the effect of Si fertilization on the uptake and transport S in plants.

The aim is very interesting and important.

The introduction is clear and logic.

The material and method is described the experiment very well.

The obtained data is clear and discussion is very good.

All presented tables and figures is clear as well.

The manuscript can be published without any remarks.

Author Response

Please find our response in the joined file.

Reviewer 2 Report

The manuscript entitled “Assessing the effect of silicon supply on root sulfur uptake in 2

S-fed or S-deprived Brassica napus L.”  submitted by Laine et al., focuses on how Si improves root sulfate uptake capacity and alleviates the disorder induced by S-deficiency in brassica napus as it is called the S-hunger plant.  Authors compiled the database comprehensibly and applied methods are novel to achieve the objective of this study. The introduction provides a good understanding of the subject and its importance, with a significant quantity of information.    However, I suggest authors include some key references related to this study in introduction and discussion section sections

# https://doi.org/10.1093/jxb/eraa291

# https://doi.org/10.1016/j.plaphy.2020.09.038

# https://doi.org/10.1007/s10725-021-00787-5

# DOI: 10.1016/j.niox.2019.11.002

#DOI: 10.1016/j.ecoenv.2019.05.043

# DOI: 10.1007/s00709-011-0273-6

# In Figure 1: The authors have mentioned that “Different Greek and Latin lowercase letters indicate that the mean dry weights of whole plants and each plant compartment (roots and shoots) are significantly different (p< 0.05), respectively”.  Better to write separately I think  Greek letters represent SE or SD and Latin letters significantly different.  Also, I suggest authors to check it carefully because the last two Bars (-S-Si and -S+Si) have b and bc, how. Please check the statistical analysis.

#  Authors should maintain uniformity in all Figures presentation and also in footnotes writing. Figure 2 footnotes and presentation are ok, not confusing,   I suggest authors exclude Greek letters in all figures. Latin letters are enough.  

#How authors were decided that 500 μM  of S and 1.7 mM  of Si are optimum concertations for Brassica napus, please include in M&M 

Author Response

Please find our responses in the joined file.

Reviewer 3 Report

The manuscript by Laine et al. is focused on an interesting topic of Si interactions with nutrients, i.e. effect of Si on S uptake along with the expression of sulfate transporter genes (Sultr 1;1 and Sultr 1;2). The study is generally well conducted and the presented data are sound. The first impression of the study is nice, however after careful reading one could find some aspects that need to be clarified: 

1) The main "take-home message" is that 
Si supply did not mitigate overall growth performance and S uptake of S deficient rapeseed plants and even down-regulated the expression of BnaSultr1;1. I thus concluded that Si does not mitigate S deficiency to the extent of for instance P deficiency in grains (see Kostic et al. 2017. Plant Soil 419, 447-455).

2) It is a bit confusing description of 34S determination and calculation of the 34S amount (including equations).

3) Week point of this work is using of Na-silicate without conversion to Si(OH)4 as information about pH is missing. Thus, more detail on control of pH and soluble Si in nutrient solutions must be included in Materials & Methods, as Si affects solution pH in a very significant way and can cause strong pH drifts over time. Note that a lower concentration of silicic acid (than you desired) in the nutrient solution might explain the diminished effect of Si.  

4) The only clear effect of Si is mitigation of S-deficiency-induced oxidative stress due to increased accumulation of phenolics (e.g. anthocyanins).

Minor queries:

1. 
1.7 mM Si means what? Did you check the real Si concentration in the nutrient solution? 

2. Correct typos in line 370

3. Total Si and S amounts in Fig. 3 should be presented separately for root and shoot. 

Author Response

Please find our responses in joined file
